# Influencing Cardiovascular Outcomes through Heart Rate Variability Modulation: A Systematic Review

**DOI:** 10.3390/diagnostics11122198

**Published:** 2021-11-25

**Authors:** Alexandru Burlacu, Crischentian Brinza, Iolanda Valentina Popa, Adrian Covic, Mariana Floria

**Affiliations:** 1Faculty of Medicine, University of Medicine and Pharmacy, 700115 Iasi, Romania; alexandru.burlacu@umfiasi.ro (A.B.); crischentian.brinza@d.umfiasi.ro (C.B.); adrian.covic@umfiasi.ro (A.C.); 2Institute of Cardiovascular Diseases, 700503 Iasi, Romania; 3Medical Sciences Academy, 700503 Iasi, Romania; 4Nephrology Clinic, Dialysis, and Renal Transplant Center, University Hospital, 700503 Iasi, Romania; 5Military Emergency Clinical Hospital, 700483 Iasi, Romania

**Keywords:** heart rate variability, biofeedback, cardiovascular diseases, cardiovascular outcomes, breathing

## Abstract

Psychological stress is a well-established risk factor for cardiovascular disease (CVD). Heart rate variability (HRV)-biofeedback could significantly reduce stress levels and improve autonomic nervous system function and cardiovascular endpoints. We aimed to systematically review the literature to investigate the impact of HRV modulation through HRV-biofeedback on clinical outcomes in patients with CVD. A literature search was performed in the following databases: MEDLINE (PubMed), Embase, and Cochrane from the inception until 1 October 2021. Patients in the HRV-biofeedback group had significantly lower rates of all-cause readmissions than patients who received psychological education (respectively, *p* = 0.028 and *p* = 0.001). Heart failure following HRV-biofeedback displayed an inverse association with stress and depression (respectively, *p* = 0.022 and *p* = 0.033). When stratified according to left ventricular ejection fraction (LVEF), patients with LVEF ≥ 31% showed improved values of the 6 min walk test after HRV-biofeedback interventions (*p* = 0.05). A reduction in systolic and diastolic blood pressure associated with HRV-biofeedback was observed (*p* < 0.01) in pre-hypertensive patients. HRV-biofeedback had beneficial effects on different cardiovascular diseases documented in clinical trials, such as arterial hypertension, heart failure, and coronary artery disease. A standard breathing protocol should be applied in future studies to obtain equivalent results and outcomes. However, data regarding mortality in patients with coronary artery disease are scarce and need further research.

## 1. Introduction

Heart rate variability (HRV) represents a non-invasive, indirect parameter of autonomic nervous system function. It reflects the fluctuation in time of successive sinus heartbeats related to the balance between parasympathetic and sympathetic nervous systems. Solid evidence presented HRV as a surrogate correlative marker of the deep, complex interaction between the nervous system (brain) and the heart rhythm [1,2,3].

The interest in HRV assessment has increased in the last few years. Although novel technologies of HRV measurement have been developed, including different wearable devices, guidelines and recommendations from the Task Force of The European Society of Cardiology (ESC) and the North American Society of Pacing and Electrophysiology have not been updated since 1996 [4]. HRV is not measured as a single value, but rather as a variety of time- and frequency-domain parameters, as well as non-linear methods of measurement. Among time-domain parameters, the most used in clinical trials are the standard deviation of all NN intervals (SDNN), the standard deviation of the average NN interval over short time divisions (SDANN), HRV triangular index, and the square root of the mean squared differences of consecutive NN intervals (RMSSD), which are endorsed by the existing guidelines [4]. Power in the low-frequency range (LF), power in the high-frequency range (HF), and the ratio between LF and HF are usually used as frequency-domain methods to characterize HRV [4].

The utility of HRV measurement is based on recent studies involving patients with cardiovascular, lung, or chronic kidney diseases and different conditions affecting the autonomic nervous system and brain activities (emotional state, stress level, fatigue, sleepiness, alert state) [5,6,7,8,9]. HRV assessment helps identify patients with a higher risk of all-cause death and adverse cardiovascular events, as documented in a solid meta-analysis [8].

Fascinating implications of HRV measurement are represented by psychological stress level monitoring and management, as it is a well-established risk factor for cardiovascular disease, including atherosclerotic cardiovascular disease (ASCVD) [10,11]. One meta-analysis concluded that HRV could be used for psychological stress evaluation. In stressful conditions, the lower parasympathetic activity is translated into reduced HF and increased LF bands [7].

Recent studies documented that HRV is not just a correlative marker but could also be modulated to improve clinical outcomes. HRV-biofeedback, a slow breathing technique (approximatively six breaths/minute), could improve HRV parameters and vagal tone. In a recent systematic review, HRV-biofeedback could benefit patients with various chronic pathological conditions, including hypertension, asthma, inflammation, depression, anxiety, sleep disorders, and pain [12]. In addition, another meta-analysis confirmed that HRV-biofeedback had beneficial effects on depressive symptoms, suggesting its utility in improving psychological health [13].

The 2021 ESC guidelines on cardiovascular disease prevention advocate psychotherapeutic stress management to improve cardiovascular outcomes and reduce stress symptoms in ASCVD patients (class IIa recommendation, level of evidence B) [14]. However, this recommendation was based on studies focused on education and cognitive-behaviour therapies to reduce stress, but not on the HRV-biofeedback technique. Nevertheless, HRV-biofeedback could significantly reduce stress levels and improve autonomic nervous system function and cardiovascular endpoints [13,15].

We aimed to systematically review the literature to investigate the impact of HRV modulation through HRV-biofeedback on clinical outcomes in patients with cardiovascular diseases.

## 2. Materials and Methods

We conducted the present systematic review in concordance with updated Reporting Items for Systematic Review and Meta-Analyses (PRISMA) guidelines [16]. The protocol was registered in PROSPERO (CRD42021286471).

### 2.1. Data Sources and Search Strategy

A literature search was performed in the following databases: MEDLINE (PubMed), Embase, and Cochrane from the inception until 1 October 2021. Time intervals and language filters were not applied. Additionally, the search was restricted to studies that enrolled humans. In order to find additional citations which could be included in the analysis, the search was extended to Google Scholar and ClinicalTrials.gov databases. References from relevant studies were also screened for additional citations. We used several prespecified MeSH terms and keywords in the search process in above mentioned databases: “heart rate variability”, “biofeedback”, “resonance frequency breathing”, “cardiovascular events”, “mortality”, “heart failure”, “coronary artery disease”, “arterial hypertension”. In line with PRISMA recommendations, full search strategies in all databases were provided in Appendix A.

### 2.2. Eligibility Criteria and Outcomes

Several essential inclusion criteria were defined and applied for retrieved citations which guided the eligibility assessment: (1) randomized and non-randomized studies which enrolled at least ten humans aged 18 years or more; (2) patients with known cardiovascular disease were investigated; (3) studies that analysed HRV-biofeedback as the intervention group and other active interventions or no-intervention as comparator group (when available); (4) studies which reported original data regarding the impact of HRV-biofeedback on clinical outcomes in patients with cardiovascular diseases (all-cause and cardiac mortality, all-cause readmissions, the 6 min walk test, blood pressure, heart rate, quality of life, stress, and depression). Additionally, studies were excluded from the analysis if they met at least one of the following criteria: studies available only in abstracts, editorials, letters, case reports, overlapping population, conference papers, unpublished data, meta-analyses, and inability to extract data. Two independent investigators decided if studies fulfilled the inclusion criteria, and any disagreements were solved by discussion and consensus.

### 2.3. Data Collection

Two independent investigators extracted the following data from each included study: the first author, design of the study and publication year, number of patients enrolled and their age, clinical setting, type of intervention, comparator group (when available), investigated outcomes, and follow-up duration. Whenever possible, data were reported as numbers, intervals, percentages, risk ratio (RR), median and mean values, confidence intervals, and *p* values.

### 2.4. Quality and Risk of Bias Assessment

The risk of bias of randomized trials was assessed using the revised Cochrane risk-of-bias tool for randomized trials (RoB 2). This tool encompasses essential signalling questions grouped in trial design, conduct, and reporting domains [17]. The quality of observational non-randomized studies was appraised using the Newcastle–Ottawa scale. It contains a series of questions grouped in three domains: selection, comparability of groups, and the investigated outcomes [18].

## 3. Results

We searched the databases mentioned above and retrieved 452 references. Finally, 53 articles were assessed for eligibility criteria after excluding duplicate citations and those based on title or abstract. Further, additional references were excluded, namely studies available only in abstract and those which did not meet the inclusion criteria, leaving 12 studies for inclusion in our systematic review. The flow-chart of the search process as well as complete databases search strategies were provided in Figure 1 and Appendix A, respectively.

Demographic and clinical data, as well as general characteristics of studies analysed, were presented in Table 1.

Additionally, results and outcomes reported in clinical studies were provided in Table 2.

Notably, most studies were randomized controlled trials [19,20,21,22,23,24,25,26,27], while only three studies had an observational design [28,29,30]. Additionally, only two studies were performed in multiple centres [19,21]. Regarding follow-up duration, only one study reported 1-year outcomes, while the rest had a shorter follow-up, ranging from 4 weeks to 18 weeks [19]. Concerning clinical settings, studies investigated mainly the role of HRV-biofeedback in patients with arterial hypertension or “pre-hypertension” [22,23,24,26,27,29,30], followed by coronary artery disease [19,21,25] and heart failure patients [20,28].

Only one study investigated the 1-year risk of all-cause readmissions and emergency visits in patients with coronary artery disease [19]. Yu et al. revealed that patients randomized to the HRV-biofeedback group had a significantly lower rate of all-cause readmissions and emergency visits than patients from the control group who received psychological education (respectively, *p* = 0.028 and *p* = 0.001). These findings were maintained after multivariate analysis. Although patients in the HRV-biofeedback group tended to have a lower rate of cardiac emergency visits than patients from the control group (respectively, 4.00% and 5.08%), it did not reach statistical significance. Interestingly, patients allocated to the HRV-biofeedback group exhibited a decrease in the Beck Depression Inventory score, while the control group experienced an increase in the total score. The authors also aimed to research the impact of HRV-biofeedback on 1-year mortality, but no deaths occurred during the follow-up period [19].

Nolan et al. observed similar neuro-psychological effects of HRV-biofeedback in the case of patients with coronary heart disease [21]. Although the intervention was performed during a short time frame (4 weeks), HF following HRV-biofeedback displayed an inverse association with stress and depression (respectively, *p* = 0.022 and *p* = 0.033). Contrary to patients allocated to the HRV-biofeedback group, there was no documented association with stress and depression in the case of patients from the active control group [21].

In another randomized controlled trial involving patients with coronary artery disease, Climov et al. found discrepant results compared to the studies mentioned above [25]. Although depression and anxiety scores tended to be lower in HRV-biofeedback patients, the difference did not reach statistical significance. In addition, both systolic and diastolic blood pressure were similar before and after the intervention. Additionally, no difference was observed in blood pressure levels between the HRV-biofeedback and control groups. However, the study did not enrol specifically hypertensive subjects who could have different results. Moreover, Climov et al. and Nolan et al. included a small number of patients, thus limiting the results [21,25].

HRV-biofeedback could improve exercise tolerance, in patients with heart failure, as observed by Swanson et al. [20]. When stratified according to left ventricular ejection fraction (LVEF), patients with LVEF ≥ 31% showed improved values of the 6 min walk test after HRV-biofeedback interventions (*p* = 0.05). However, the quality of life remained similar before and after HRV-biofeedback intervention. Nonetheless, results are limited by the small sample size, and these findings should be analysed in more extensive trials [20].

HRV-biofeedback could have beneficial effects on blood pressure and baroreflex sensitivity in patients with chronic heart failure. Bernardi et al. observed a reduction in systolic and diastolic blood pressure in heart failure patients following HRV-biofeedback (respectively, *p* = 0.009 and *p* = 0.02). Additionally, slow breathing was associated with increased baroreflex sensitivity in both heart failure patients and healthy controls (*p* < 0.0025) [28].

Although studies investigating the impact of HRV-biofeedback on blood pressure included a small number of patients, they documented similar results. Lin et al. included asymptomatic prehypertensive patients and observed a reduction in systolic and diastolic blood pressure associated with HRV-biofeedback (*p* < 0.01). Additionally, when compared to slow abdominal breathing, HRV-biofeedback decreased significantly systolic blood pressure (*p* < 0.05). In addition, HRV-biofeedback improved baroreflex sensitivity (*p* < 0.01) [22]. In another study involving prehypertensive patients, Chen et al. documented that HRV-biofeedback reduced systolic and diastolic blood pressure as compared to slow abdominal breathing (*p* < 0.05) [26].

In addition to resting systolic blood pressure levels, Jones et al. observed that HRV-biofeedback was associated with a reduced systolic in response to exercise (*p* < 0.05) [23,24]. Additionally, slow breathing training reduced resting heart rate, as we all heart rate in response to exercise (*p* < 0.05 for both) [23,24].

Albuquerque Cacique et al., and Nolan et al., enrolled hypertensive patients where HRV-biofeedback was linked to systolic blood pressure reduction, but had an insignificant effect on diastolic blood pressure levels [27,29]. The latter study also documented a decreased number of patients with uncontrolled blood pressure following slow breathing training, with a number needed to treat of 7 [27]. Similar results were found by Joseph et al., namely a reduction in systolic and diastolic blood pressure in hypertensive patients linked to slow breathing technique [30].

The risk of bias of randomized trials assessed using the RoB 2 tool [17] and quality appraisal using the Newcastle–Ottawa scale [18] is provided in Appendix A and Appendix A, respectively.

## 4. Discussion

To the best of our knowledge, the present systematic review is the first one to investigate the impact of HRV-biofeedback, a slow breathing technique, on various clinical outcomes in patients with cardiovascular diseases (Figure 2).

Raising HRV through breathing techniques would therefore influence cardiovascular outcomes (Table 1). The current studies included in our systematic review, however, did not have as major objectives cardiovascular mortality, but only “soft” endpoints (Table 2).

The idea of HRV modulation emerged due to increasing evidence for a correlation between HRV and adverse events in various conditions. A lower HRV was associated with a more than two-fold higher risk of all-cause death and cardiovascular events in patients with cardiovascular diseases. Additionally, in patients with acute myocardial infarction, HRV was associated with a significantly higher risk of all-cause mortality [8]. Thus, self-regulation intervention on HRV could improve outcomes of patients with cardiovascular diseases.

HRV measurement and HRV-biofeedback could be efficiently implemented in daily clinical practice and life once different wrist-worn devices became available. Some existing devices received CE-mark for medical use, including HRV measurement and arrhythmia monitoring [31,32]. In addition, initial patients’ education could be enough for further home HRV-biofeedback training using wrist-worn devices, which could guide the respiration process.

HRV-biofeedback implies breathing at a slower rate, close to the resonance frequency, which causes the highest fluctuations of heart rate and baroreflex stimulation. However, the protocols used in analysed studies were different in terms of breathing rate, the number of breathing cycles, the number of sessions per week, and intervention duration. The clinical studies’ most used breathing rate was six breaths/minute, but it could be located between 4.5 and 6.5 breaths/minute. Thus, more studies are required to evaluate the superiority of breathing at an individual resonance frequency over slow breathing at a standard rate (6 breaths/minute) [33]. Notably, a standard HRV-biofeedback protocol is essential for future clinical trials to obtain equivalent results and outcomes. In this regard, authors from one study proposed a protocol for HRV-biofeedback training based on their previous research in the field. The authors described a new 5-visit protocol (revised from the last 10-session protocol), including recommendations for resonance frequency ascertainment. Additionally, it highlighted the importance of long-term home practice to maintain baroreflex trained [34].

A primary mechanism involved in HRV-biofeedback is represented by parasympathetic nervous system stimulation, which attenuates the effects of sympathetic nervous system activation. Therefore, HRV-biofeedback could have similar effects to beta-blockers, potentially beneficial in patients with arterial hypertension, heart failure, or coronary artery disease [35,36].

Arterial hypertension is characterized by autonomic nervous system dysfunction, with increased sympathetic activity. Sympathetic overdrive promotes the progression of vascular and cardiac complications of the disease [37]. Recent solid evidence demonstrated that hypertensive patients exhibited lower HRV values and baroreflex sensitivity due to arterial stiffness [38]. All these data suggest the possible utility of HRV-biofeedback in patients with arterial hypertension.

Analysed studies from our systematic review displayed concordant results regarding HRV modulation in hypertensive patients, except one study, which did not reveal any significant effect linked to HRV-biofeedback [25]. One study documented that HRV-biofeedback improved autonomic nervous system markers since HF increased compared to active controls [27]. Similar findings with increased HF and lower LF/HF ratio following HRV-biofeedback training were observed in another study [23]. In addition, one study documented increased both HF and LF after HRV-biofeedback. However, patients randomized to slow abdominal breathing exhibited an increase only in LF, with no effect on HF, suggesting an improvement of parasympathetic activity associated with HRV-biofeedback, but not with slow abdominal breathing [22]. These data pointed out the parasympathetic stimulation as one of the possible mechanisms of blood pressure decrease following HRV-biofeedback. Although most included studies were randomized, they enrolled a small number of patients, thus limiting the results to larger populations.

Patients with heart failure are also marked by autonomic nervous system dysfunction. Heart failure is characterized by increased sympathetic function and decreased parasympathetic activity. This imbalance leads to further cardiac remodelling and structural changes. There were described non-pharmacological invasive and less-invasive methods to diminish the sympathetic activity, such as renal nerve ablation, vagus nerve activity stimulation, and different electrical stimulation [39].

Thus, HRV-biofeedback is of particular interest in this setting, as it is a relatively simple non-invasive technique that could be implemented in cardiac rehabilitation programs. Existing studies focused on exercise tolerance and blood pressure reduction in heart failure patients [20,28]. Thus, future trials are awaited to investigate the impact of HRV-biofeedback on mortality and adverse cardiovascular events, especially in the context of current optimal medical therapy.

Finally, HRV-biofeedback appeared to have beneficial effects in the case of patients with coronary artery disease in several ways: all-cause readmission, all-cause emergency visits, depression, and stress reduction [19,21]. However, data regarding mortality in patients with coronary artery disease are scarce and need further research.

## 5. Conclusions

HRV-biofeedback had beneficial effects on different cardiovascular diseases documented in clinical trials, such as arterial hypertension, heart failure, and coronary artery disease. Autonomic nervous system activity improvement represents one of the essential mechanisms through which HRV-biofeedback influences cardiovascular outcomes. A standard breathing protocol should be applied in future studies to obtain equivalent results and outcomes. More research is required to investigate the impact of HRV-biofeedback on mortality and adverse cardiovascular events in the context of contemporary patients with cardiovascular diseases.

## Figures and Tables

**Figure 1 diagnostics-11-02198-f001:**
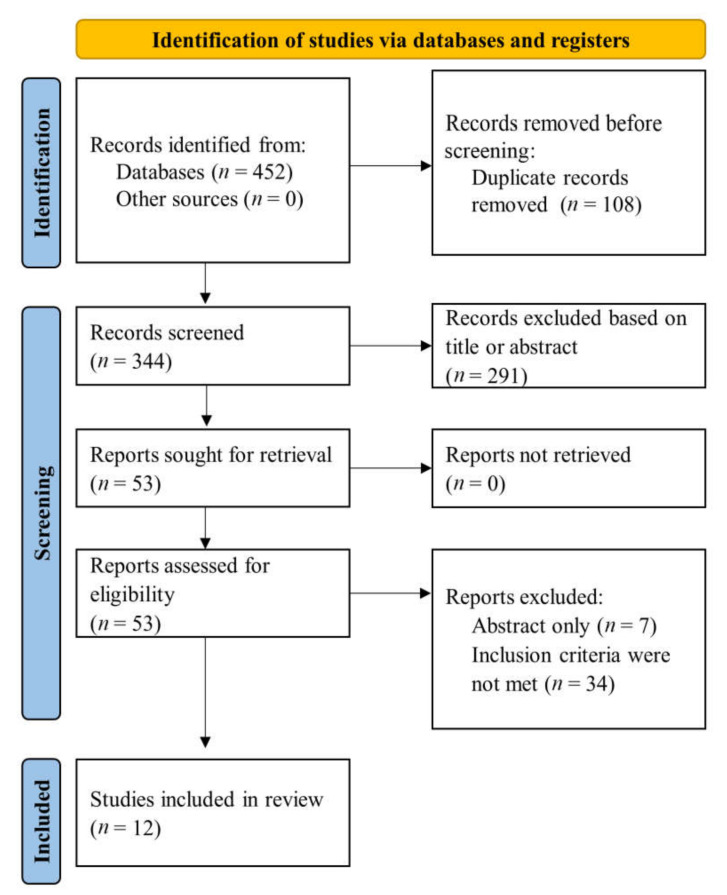
Flow diagram of selected studies in present systematic review.

**Figure 2 diagnostics-11-02198-f002:**
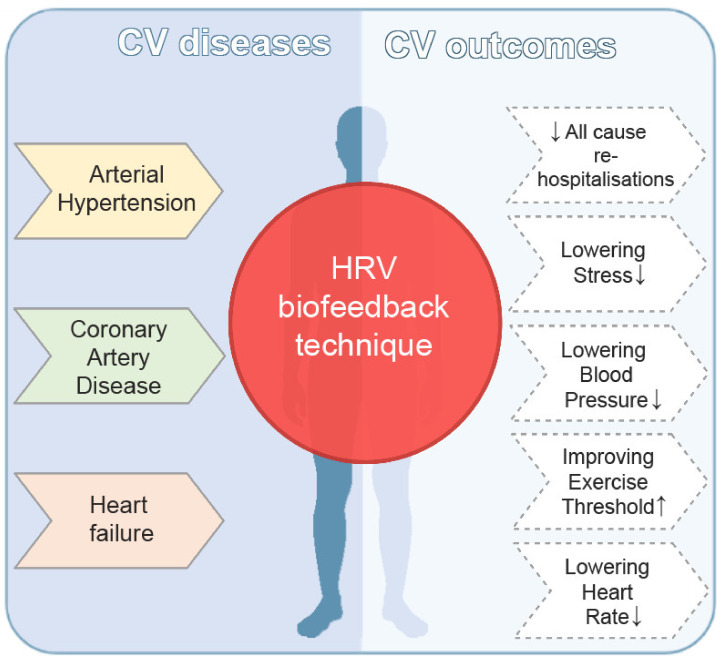
Various clinical outcomes of HRV-biofeedback technique in patients with cardiovascular diseases.

**Table 1 diagnostics-11-02198-t001:** General characteristics of studies included in present systematic review.

Author, Year	Design	Patients, No.	Age, Mean/ Median ± SD	Setting	Intervention and Comparator	Outcomes	Follow-Up
Yu et al., 2018	Randomized, controlled, single-blinded, multicentre	210	61.24—HRV-biofeedback group	Patients with coronary artery disease	HRV-biofeedback group (*n* = 105) versus control group (*n* = 105) which included 10 min of psychological education	(a) all-cause and cardiac readmissions (b) emergency revisits (c) mortality (d) all-cause events	1 year
			60.31—control group				
Swanson et al., 2009	Randomized, controlled, single-blind, single centre	29	54 ± 11—HRV-biofeedback group	Patients with NYHA class I–III heart failure	HRV-biofeedback (*n* = 15; once per week for 45 min at weeks 1–6) versus control group (*n* = 14)	(a) exercise tolerance (the 6 min walk test) (b) quality of life (LHFQ)	18 weeks
			56.4 ± 13.5—control group				
Nolan et al., 2005	Randomized, controlled, multicentre	46	54.22 ± 1.04—HRV-biofeedback group	Patients with coronary heart disease (myocardial infarction, or positive diagnostic test)	HRV-biofeedback, 5 sessions, 6 breaths/minute (*n* = 27) versus active control group (*n* = 19)	(a) symptoms of depression (CES-D) (b) symptoms of psychological stress (PSS)	4 weeks
			54.95 ± 1.52—active control group				
Lin et al., 2012	Randomized, controlled, single centre	43	22.3	Asymptomatic patients with prehypertension (systolic blood pressure = 120-139 mmHg and/or diastolic blood pressure = 80–89 mmHg)	HRV-biofeedback group (*n* = 18, 10 sessions over 5 weeks), slow abdominal breathing group (*n* = 15) and control group (*n* = 10)	(a) Systolic blood pressure (b) baroreflex sensitivity	3 months
Jones et al., 2015	Randomized, controlled, single centre	30	53.4 ± 4.3—no load training group	Patients with essential hypertension stage I–II	Slow breathing training for 8 weeks (*n* = 20, 10 with unloaded slow breathing and 10 against an inspiratory load of 20 cmH_2_O) or untrained controls (*n* = 10)	Systolic blood pressure and heart rate response to handgrip exercise	10 weeks
			51.4 ± 5.3—loaded training group				
			50.4 ± 5.4—control group				
Jones et al., 2010	Randomized, controlled, single centre	30	53 ± 4—no load training group	Patients with essential hypertension stage I–II	Slow deep breathing at home, 30 min sessions, twice daily for 8 weeks (*n* = 20, 10 with unloaded breathing and 10 against an inspiratory load of 20 cm H_2_O) or control group (*n* = 10, normal daily life)	Resting blood pressure and heart rate (measured at home and in the laboratory)	9 weeks
			51 ± 5—loaded training group				
			50 ± 5— control group				
Climov et al., 2014	Randomized, controlled, single centre	31	61.3 ± 6.2—HRV- biofeedback group	Patients with coronary artery disease	HRV-biofeedback group (10 sessions of 45–60 min in addition to rehabilitation programme) versus a control group in a cardiac rehabilitation centre setting	(a) systolic and diastolic blood pressure (b) anxiety and depression	6 weeks
			51.8 ± 9.7—control group				
Chen et al., 2015	Randomized, controlled, single centre	32	21.5 ± 0.18	Prehypertensive patients (systolic blood pressure = 120–139 mmHg and/or diastolic blood pressure = 80–89 mmHg)	HRV-biofeedback group (*n* = 12, 15 sessions), slow abdominal breathing group (*n* = 10) or control group (*n* = 10)	(a) heart rate (b) blood pressure (c) blood volume pulse amplitude	3 months
Bernardi et al., 2002	Observational	102	58 ± 1—chronic heart failure group	Patients with stable chronic heart failure, NYHA classes I–IV	Slow breathing rate (4 min of controlled breathing 15/min and 4 min of controlled breathing 6/min) or spontaneous breathing	(a) baroreflex sensitivity (b) blood pressure	–
			55 ± 2— control group				
Albuquerque Cacique et al., 2021	Observational, single centre	16	57.3 ± 14	Patients with essential hypertension stage I–II	Biofeedback paced breathing, 8 sessions for 20 min	(a) systolic and diastolic blood pressure (b) anxiety (Hamilton anxiety rating scale) (c) stress (PSS)	–
Nolan et al., 2010	Randomized, controlled	65	55.0 ± 1.2—HRV- biofeedback group	Patients with uncomplicated arterial hypertension	HRV-biofeedback (6 breaths/minute or active control (autogenic relaxation) —six 1 h sessions	(a) daytime systolic and diastolic blood pressure, and pulse pressure (b) 24 h systolic and diastolic blood pressure, and pulse pressure	9 weeks
			55.9 ± 1.2— active control group				
Joseph et al., 2005	Observational	46	56.4 ± 1.9— hypertensive patients	Patients with essential hypertension (*n* = 20) and healthy controls (*n* = 26)	Slow breathing (6 breaths/minute) or spontaneous breathing or faster breathing (15 breaths/minute)	(a) systolic and diastolic blood pressure (b) baroreflex sensitivity	–
			52.3 ± 1.4—healthy controls				

CES-D = Centre for Epidemiologic Studies in Depression scale; HRV = heart rate variability; LHFQ = Minnesota Living with Heart Failure Questionnaire; NYHA = New York Heart Association; PSS = Perceived Stress Scale.

**Table 2 diagnostics-11-02198-t002:** Results reported in clinical studies included in present systematic review.

Author, Year	Outcomes	Results	
Yu et al., 2018		HRV-BF Group vs. Control Group	
	All-cause readmissions	12.0% vs. 25.42% (RR = 0.31, 95% CI, 0.11–0.84)	*p* = 0.028
	All-cause emergency visits	13.33% vs. 35.59% (RR = 0.26, 95% CI, 0.11–0.63)	*p* = 0.001
	All-cause and cardiac mortality	No deaths were reported	
Swanson et al., 2009		HRV-BF Group vs. Control Group	
	The 6 min walk test (Patients with LVEF ≥ 31%)	Baseline: 432 ± 77 m vs. 416 ± 166 m; Follow-up: 485 ± 109 m vs. 385 ± 160 m	*p* = 0.05
	The 6 min walk test (Patients with LVEF ≤ 30%)	Baseline: 394 ± 73 m vs. 318 ± 113 m; Follow-up: 395 ± 87 m vs. 325 ± 115 m	
	LHFQ (patients with LVEF ≥ 31%)	Baseline: 33.0 ± 23.2 m vs. 33.7 ± 15.9 m; Follow-up: 38.0 ± 19.5 m vs. 22.2 ± 23.3 m (post-intervention)	*p* = 0.66
Nolan et al., 2005	Stress—HRV-BF group (logHF)	Adjusted R^2^ = 0.86	*p* = 0.022
	Depression—HRV-BF group (logHF)	Adjusted R^2^ = 0.81	*p* = 0.033
	Stress—active control group (logHF)	Adjusted R^2^ = 0.04	*p* = 0.567
	Depression—active control group (logHF)	Adjusted R^2^ = 0.13	*p* = 0.946
Lin et al., 2012	Blood pressure	Baseline: 131.7 ± 8.7/79.3 ± 4.7 mmHg After intervention: 118.9 ± 7.3/71.9 ± 4.9 mmHg 3 months: 118.9 ± 6.6/72.4 ± 5.6 mmHg	*p* < 0.01
	Systolic blood pressure	HRV-BF vs. Slow Abdominal Breathing	*p* < 0.05
	Baroreflex sensitivity	Baseline: 7.0 ± 5.9 ms/mmHg After intervention: 15.8 ± 5.3 ms/mmHg	*p* < 0.01
Jones et al., 2015	Systolic blood pressure in response to exercise	After slow breathing training, systolic blood pressure response was reduced by 10 mmHg (95% CI, −7 to −13)	*p* < 0.05
	Heart rate in response to exercise	After slow breathing training, heart rate response was reduced by 5 beats per minute (95% CI, −4 to −6)	*p* < 0.05
Jones et al., 2010	Resting systolic blood pressure	Decreased with 7.0 mmHg (95% CI, 5.5–8.5) in unloaded breathing group and with 18.8 mmHg (95% CI, 16.1–21.5) in loaded breathing group compared to control group	*p* < 0.05
	Resting heart rate	Decreased with 8 beats/minute (95% CI, 6.5–10.3) in unloaded breathing group and with 9 beats/minute (95% CI, 5.6–12.2) in loaded breathing group	*p* < 0.05
Climov et al., 2014	Systolic blood pressure	No statistically significant differences between the two groups	*p* = 0.64
	Diastolic blood pressure		*p* = 0.34
	Depression and anxiety		
Chen et al., 2015	Systolic blood pressure	Decreased from 131.58 ± 8.41 mmHg to 116.17 ± 9.25 mmHg in HRV-BF group vs. control group	*p* < 0.01
	Diastolic blood pressure	Decreased from 81.33 ± 3.06 mmHg to 71.17 ± 7.12 mmHg in HRV-BF vs. control group	
	Systolic and diastolic blood pressure	Decreased significantly in HRV-BF compared to slow abdominal breathing group at 3 months	*p* < 0.05
Bernardi et al., 2002	Systolic blood pressure	Decreased from 117 ± 3 mmHg to 110 ± 4 mmHg	*p* = 0.009
	Diastolic blood pressure	Decreased from 62 ± 1 mmHg to 59 ± 1 mmHg	*p* = 0.02
	Baroreflex sensitivity	Increased from 5.0 ± 0.3 ms/mmHg to 6.1 ± 0.5 ms/mmHg (in chronic heart failure patients) and from 9.4 ± 0.7 ms/mmHg to 13.8 ± 1.0 ms/mmHg (in healthy controls)	*p* < 0.0025
Albuquerque Cacique et al., 2021	Systolic blood pressure	Decreased from 120 ± 16 mmHg to 111 ± 21 mmHg	*p* = 0.002
	Diastolic blood pressure	Decreased from 74.8 ± 9 mmHg to 72.1 ± 8 mmHg	*p* = 0.13
	Anxiety (Hamilton anxiety rating scale)	Decreased from 17.2 ± 9 to 11 ± 7	*p* = 0.0009
	Stress (PSS)	Decreased from 15 ± 10 to 13 ± 5	*p* = 0.37
Nolan et al., 2010	Systolic blood pressure reduction	Daytime: −2.4 ± 0.9 mmHg 24 h: −2.1 ± 0.9 mmHg	*p* = 0.009 *p* = 0.03
	Pulse pressure reduction	Daytime: −1.7 ± 0.6 mmHg 24 h: −1.4 ± 0.6 mmHg	*p* = 0.004 *p* = 0.02
	Diastolic blood pressure	Daytime and 24 h	*p* > 0.10
	Uncontrolled blood pressure	Number of patients decreased from 17 (pre-treatment) to 12 after HRV-BF with a number needed to treat = 7 (95% CI, 4–57)	
Joseph et al., 2005	Systolic blood pressure	Decreased from 149.7 ± 3.7 mmHg to 141.1 ± 4 mmHg	*p* < 0.05
	Diastolic blood pressure	Decreased from 82.7 ± 3 mmHg to 77.8 ± 3.7 mmHg	*p* < 0.01
	Baroreflex sensitivity	Increased from 5.8 ± 0.7 ms/mmHg to 10.3 ± 2.0 ms/mmHg	*p* < 0.01

HF = power in high frequency range; HRV = heart rate variability; HRV-BF = heart rate variability biofeedback; LHFQ = Minnesota Living with Heart Failure Questionnaire; LVEF = left ventricular ejection fraction; PSS = Perceived Stress Scale; RR = relative risk.

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
