# Peer review of "Influencing Cardiovascular Outcomes through Heart Rate Variability Modulation: A Systematic Review"

_diagnostics, 2021, doi:10.3390/diagnostics11122198_

Round 1
Reviewer 1 Report
The review is well written and the methodological approach is correct. I have only a few suggestions:
- the recent technological advent of the last decades introduces wrist-worn devices that permit recording RR data during daily life. I think that could be interesting to add some description about these devices and their utility in assessing the cardiac health status of the population.
- I think that it is better to explain in the introduction and results sections the protocol of HRV-biofeedback intervention. Did the studies perform the same protocol? If not, please describe the different protocols.
Reviewer 2 Report
In the present manuscript, authors investigate a very interesting issue in cardiovascular diseases: the role of heart rate variability in hypertension, CAD and HF. The analysis is well performed and give very interesting feedback for application in clinical practice. The only major issue is the routine use in the real world because a specific training should be necesessary. Thus, I would like to suggest the authors to include in their discussion all the potential warnings for the use in clinical practice
